# Characterization of occult hepatitis B in high-risk populations in Kenya

Kiptoon Beatrice Jepkemei [1¤a], Missiani Ochwoto[2], Ken Swidinsky[1], Jacqueline Day[1], Henok Gebrebrhan[3], Lyle R. McKinnon[3,4,5], Anton Andonov[1¤b], Julius Oyugi[3,4], Joshua Kimani[3,4], George Gachara [6], Elijah Maritim Songok[2,3], Carla Osiowy [1,3]*

1 National Microbiology Laboratory, Public Health Agency of Canada, Winnipeg, Manitoba, Canada, 2 Kenya Medical Research Institute, Nairobi, Kenya, 3 Department of Medical Microbiology and Infectious Diseases, University of Manitoba, Winnipeg, Manitoba, Canada, 4 Department of Medical Microbiology, University of Nairobi, Nairobi, Kenya, 5 Centre for the AIDS Programme of Research in South Africa (CAPRISA), Durban, South Africa, 6 Department of Medical Laboratory Sciences, Kenyatta University, Nairobi, Kenya

¤a Current address: Department of Medical Laboratory Sciences, Kenyatta University, Nairobi, Kenya
¤b Current address: Department of Medical Microbiology and Infectious Diseases, University of Manitoba, Winnipeg, Manitoba, Canada
* carla.osiowy@canada.ca

**Data Availability Statement:** All sequence files are available from the International Nucleotide Sequence Database Collaboration (GenBank accession numbers MK487133 - MK487155, MN972524).

## Abstract

Occult hepatitis B infection (OBI) is defined as the presence of hepatitis B virus (HBV) DNA in the liver or serum in the absence of detectable HBV surface antigen (HBsAg). OBI poses a risk for the development of cirrhosis and hepatocellular carcinoma. The prevalence of OBI in Kenya is unknown, thus a study was undertaken to determine the prevalence and molecular characterization of OBI in Kenyan populations at high risk of HBV infection. Sera from two Nairobi cohorts, 99 male sex workers, primarily having sex with men (MSM-SW), and 13 non-MSM men having HIV-positive partners, as well as 65 HBsAg-negative patients presenting with jaundice at Kenyan medical facilities, were tested for HBV serological markers, including HBV DNA by real-time PCR. Positive DNA samples were sequenced and MSM-SW patients were further tested for hepatitis C virus (HCV) infection. Of the 166 HBsAg-negative samples tested, 31 (18.7%; 95% confidence interval [CI] 13.5–25.3) were HBV DNA positive (i.e., occult), the majority (20/31; 64.5%) of which were HBV core protein antibody positive. HCV infection was not observed in the MSM-SW participants, although the prevalence of HBsAg positivity was 10.1% (10/99; 95% CI 5.6–17.6). HBV genotype A was predominant among study cases, including both HBsAg-positive and OBI participants, although the data suggests a non-African network transmission source among MSM-SW. The high prevalence of HBV infection among MSM-SW in Kenya suggests that screening programmes be instituted among high-risk cohorts to facilitate preventative measures, such as vaccination, and establish entry to treatment and linkage to care.

## Introduction

Infection with hepatitis B virus (HBV) in which viral surface antigen (HBsAg) is undetectable yet HBV DNA is detectable in the liver, and possibly in the serum, defines occult hepatitis B infection (OBI) [1]. Other concurrent serological markers of HBV infection, including

**Funding:** The author(s) received no specific funding for this work. Ms. Jepkemei was a recipient of a Queen Elizabeth II Diamond Jubilee Scholarship for the duration of the study. Dr. McKinnon is supported by a CIHR New Investigator Award.

**Competing interests:** The authors have declared that no competing interests exist.

antibody to the core (anti-HBc) or surface (anti-HBs) proteins are often present, but seronegative OBI in which HBV DNA is the only detectable marker of infection is also observed [2]. The clinical impact of OBI includes progression to severe liver disease, a risk of reactivation during immunosuppression and an ongoing risk of transmission in the context of misdiagnosis due to the lack of detectable HBsAg [3] OBI is more frequently found in populations at high risk of infection (HIV infected or people who inject drugs [PWID]) [4] and endemic regions of the world (≥8% prevalence) [5]. HBV is present at intermediate endemicity (2% to 7%) throughout sub-Saharan Africa, at approximately 6.1% [6] and has been suggested to be a neglected tropical disease disproportionately affecting this region [7]. Understandably then, OBI is fairly prevalent throughout Africa, ranging from approximately 7% to 50% in 'low-risk' populations comprised of blood donors or healthcare workers [8–11] to approximately 6% to 30% in HIV infected populations [12–15]. The prevalence of HBV infection in Kenya is estimated to be between 1% to 5% [16], although rates among specific high-risk populations (HIV co-infected, PWID, jaundiced patients seeking medical care) are much higher [17–19]. As the prevalence of OBI in Kenya is not known, the aim of this study was to investigate and characterize OBI in several Kenyan populations at high risk of infection; jaundiced patients seeking medical care, men having HIV-positive partners and male sex workers.

## Materials and methods

### Specimen collection

Ninety-nine specimens of archived sera from a follow-up sampling study to the 2009–2015 cohort study of Nairobi male sex workers primarily having sex with men (MSM-SW) described by McKinnon, et al. [20] were included in the study. The cohort peer referral convenience sampling investigation collected specimens from 127 participants, of which 99 had sufficient serum volume for investigation of OBI and thus was the basis for their selection in the current study. Similarly, the follow up sampling study also included 20 men, to serve as controls, who reported never engaging in receptive anal intercourse ("non-MSM men") and were in a serodiscordant relationship with an HIV-positive female partner. Thirteen specimens from the non-MSM cohort were included, based on the presence of sufficient serum volume for investigation of OBI. Demographic and behavioural characteristics of MSM-SW participants were collected during the original cohort studies, as was the HIV status of all MSM-SW and non-MSM men. All non-MSM participants were HIV negative, while 46.5% (46/99) of MSM-SW men were HIV positive. Thirty-two MSM-SW men (31 HIV negative, 1 HIV positive) had also been vaccinated with the HBV vaccine during a simulated HIV-1 vaccine feasibility trial (Kavi Institute of Clinical Research) without knowledge of their HBV status at the time of vaccination. Alanine aminotransferase levels of participants within each cohort were not available for analysis in this study. HBsAg negative archived sera from jaundiced patients seeking medical care at four select hospitals throughout Kenya: Kenyatta National Hospital (Nairobi), Moi Teaching and Referral Hospital (Eldoret), New Nyanza Provincial General Hospital (Kisumu), and Coast General Hospital (Mombasa) as described by Ochwoto, et al. [19] were also included in the study. The original study included 164 HBsAg negative specimens, 65 of which had sufficient serum volume remaining for investigation of OBI and thus was the basis for their selection in the current study. All jaundiced patients included in this study had been tested for antibody to hepatitis C virus (HCV) in the original study and were found to be negative [19].

**Table 1. Primer and probe sequences for HBV DNA detection and sequence analysis.**

| Primer or Probe name | 5'—3' sequence | Approximate genomic region[a] |
|---|---|---|
| Surface-FWD primer | TCCTCACAATACCRCAGAGT | 228–247 |
| Surface-REV primer | GATARCCAGGACAARTTGGAG | 371–351 |
| Surface real time PCR probe | AAAATTCGCAGTCCCCAACCTCCA | 306–329 |
| ENHI-FWD primer | AAGTGTTTGCTGACGCAA | 1178–1195 |
| ENHI-REV primer | GAGTTCCGCAGTATGGATC | 1281–1263 |
| ENHI real time PCR probe | CCATCRGCGCATGCGYGGAA | 1224–1243 |
| X/ENHII-FWD primer | CCGTCTGTTCCTTCTCATCTG | 1549–1569 |
| X/ENHII-REV primer | GTCCAAGAGTCCTCTTATGYAAG | 1671–1649 |
| X/ENHII real time PCR probe | TGCACTTCGCTTCACCTCTGCAC | 1580–1602 |
| HBPr134 first stage FWD | TGCTGCTATGCCTCATCTTC | 414–433 |
| HBPr135 first stage REV | CARAGACAAAAGAAAATTGG | 822–803 |
| HBPr75 nested stage FWD | CAAGGTATGTTGCCCGTTTGTCC | 455–477 |
| HBPr94 nested stage REV | GGYAWAAAGGGACTCAMGATG | 795–775 |

[a]According to GenBank Accession AY128092 nucleotide numbering.

## Serological testing

Specimens from the archived serum banks were initially tested for anti-HBc and HBsAg by electrochemiluminescence EIA followed by HBV DNA testing (S1 Fig). MSM-SW samples were also tested for the presence of antibody to HCV. All serological tests were performed using the COBAS e411 platform (Elecsys; Roche Diagnostics, Quebec, Canada).

## HBV DNA testing and phylogenetic analysis

Nucleic acid was extracted from 200 μL sera using an automated nucleic acid extraction system (NucliSENS easyMag, bioMerieux Inc, Saint-Laurent, QC) and eluted in 60 μL elution buffer. HBV DNA was initially detected by real-time PCR involving 3 concurrently probed genomic regions [21]. In brief, 15 μL of DNA extract was added to a reaction mixture including Quanti-Tect virus master mix (Qiagen, Toronto, ON), reference dye solution and a primer-probe mix (Table 1) specific for the HBsAg-coding region (nt 228–371, based on GenBank reference sequence AY128092), or the ENHI regulatory region (nt 1178–1281), or the X/ENHII genomic region (nt 1549–1671). Primers and probes were prepared by Integrated DNA Technologies (Kanata, ON) with oligonucleotide probes having a 5' 6-FAM fluorescent reporter dye and a double quencher (ZEN/3'IB) combination. Reaction mixtures were amplified using an Applied Biosystems 7500 real-time PCR system (ThermoFisher Scientific, Burlington, ON) including 50 cycles of 95˚C for 15s, 60˚C for 45s. Specimens were considered to be HBV DNA positive if at least two genomic regions were positive by real-time PCR. The real-time PCR method was validated against a panel of approximately 250 specimens having a viral load <6 IU/mL or "target not detected" as determined by the Cobas High Pure System/TaqMan HBV Test, as well as sensitive nested PCR of all DNA extracts [22, 23]. Ct cut-offs, below which the specimen was considered to be positive for HBV DNA, as detected for each genomic region were determined following validation, and were as follows: 37.77 (HBsAg-coding region), 40.82 (ENHI regulatory region), 38.33 (X/ENHII genomic region).

For amplification and sequencing of HBV DNA, 150 μL HBsAg positive or OBI positive study samples were extracted and PCR amplified using first stage and nested primers specific

for the HBsAg-coding region, as described by Stuyver, et al. [23] (HBVPr134/135 and HBVPr94/75; Table 1). Amplicons were gel purified and cycle sequenced using an Applied Biosystems 3730 XL DNA Analyzer (ThermoFisher Scientific, Burlington, ON). Sequences were aligned and trimmed using Clustal X v2.0.11 [24] and BioEdit v7.2.5 [25], respectively. Maximum likelihood analysis of the partial HBsAg-coding region (trimmed to 327 bp representing nt 458 to 784, based on GenBank accession AY128092) was performed using DIVEIN software [26] by the GTR+γ+I model determined as the most appropriate substitution model for the alignment. Phylogenetic tree construction was optimized by nearest neighbour interchange and subtree pruning and regrafting with branch support computed by the approximate likelihood-ratio test based on a Shimodaira-Hasegawa-like procedure [27]. Nucleotide sequence alignments were translated to amino acid in order to determine the presence of mutations affecting susceptibility to antiviral therapies (any variant at reverse transcriptase amino acid sites rtI169, rtV173, rtL180, rtA181, rtT184, rtV191, rtA194, rtS202, rtM204 was noted) and immune recognition of the HBsAg protein (any variant at HBsAg amino acid sites P105, T116, T118, G119, K/R122, S/T123, C124, T/I126, P127, Q129, T/N131, M133, S136, C139, T140, K141, P142, S/T143, D144, G145, E164, P178, Q181, and I195 was noted) [28, 29].

The risk of PCR environmental contamination was controlled with spatial and temporal separation of all steps (DNA extraction, PCR reaction pre-mix preparation, PCR amplification, and nested PCR analysis) as well as the inclusion of negative controls at the extraction and amplification steps.

### Ethical approval and informed consent

Ethical approval for collection and investigation of specimens from jaundiced patients seeking medical care was obtained from the Kenya Medical Research Institute's National Ethical Review committee, approval number SSC 2436 [19]. Informed consent was given by each participant or guardian through a signed consent form prior to drawing a blood sample and obtaining demographic information. Ethical approval for collection and investigation of specimens from MSM and non-MSM participants was obtained through institutional review boards at the Kenyatta National Hospital ERC and the University of Manitoba [20]. Both institutional research ethics boards approved the investigation of infectious pathogens in consenting participants, including the detection and characterization of hepatitis B and hepatitis C viruses.

### Statistical analysis

Fisher's exact test (two-tailed) was used to analyse the association of anti-HBc with OBI and the association of DNA positivity with MSM-SW demographic and behavioural characteristics. Chi-square analysis was used to determine the association of educational level with HBV DNA positivity. Confidence intervals of prevalence estimates were calculated by computing the confidence interval of a proportion by the Wilson/Brown method. All statistical analyses were performed using GraphPad prism (v8.4.1). $P$ values $< 0.05$ were considered significant.

## Results

### OBI and chronic HBV infection in Kenyan MSM-SW and non-MSM cohorts

The MSM-SW and non-MSM cohorts had not previously been tested for HBV, thus both OBI and chronic HBV infection (defined as HBsAg positivity) was assessed in these cohorts. The jaundiced cohort consisted of previously tested [19] HBsAg negative participants. The MSM-SW (n = 99) and non-MSM (n = 13) cohort participants were found to have a combined

**Table 2. HBsAg, anti-HBc antibody and HBV DNA results of study samples by cohort.**

| | MSM-SW (n = 99)[a] | | non-MSM (n = 13)[b] | | Jaundiced (n = 65)[c] | |
| --- | --- | --- | --- | --- | --- | --- |
| | anti-HBc positive (n = 35) | anti-HBc negative (n = 64) | anti-HBc positive (n = 6) | anti-HBc negative[c] (n = 7) | anti-HBc positive (n = 37) | anti-HBc negative (n = 28) |
| HBsAg pos DNA pos | 4 | 3 | 0 | 0 | - | - |
| HBsAg neg DNA pos[d] | 6 | 4 | 0 | 1 | 14 | 6 |

[a]Comprised of 89 HBsAg negative and 10 HBsAg positive men.

[b]Non-MSM participants were HIV uninfected men with HIV positive partners; comprised of 12 HBsAg negative and 1 HBsAg positive individuals.

[c]All study samples from jaundiced individuals were HBsAg negative.

[d]OBI.

anti-HBc positivity of 36.6% (41/112; 95% confidence interval [CI] 28.3–45.8). Six of 35 anti-HBc positive specimens from MSM-SW were HBsAg-positive, 4 of which were also HBV DNA positive (S1A Fig, Table 2), indicating chronic infection, while anti-HBc positive samples from non-MSM men were all HBsAg and HBV DNA negative (S1B Fig, Table 2). Sixty-four MSM-SW and 7 non-MSM anti-HBc negative samples were tested for HBV DNA and HBsAg; 4 of 64 anti-HBc negative specimens tested from MSM-SW were HBsAg positive, 3 of which were also HBV DNA positive (S1A Fig, Table 2), thus establishing a prevalence of HBsAg positive chronic infection among MSM-SW of 10.1% (10/99; 95% CI 5.6–17.6). One non-MSM individual was HBsAg positive (DNA negative, anti-HBc positive; S1B Fig). A finding of OBI was based on a positive HBV DNA signal by real-time PCR in at least two different genomic regions with samples from HBsAg negative individuals. All negative extraction and amplification controls were consistently negative, indicating control of possible environmental contamination. OBI was observed in 1 non-MSM and 10 MSM-SW HBsAg negative men for an OBI prevalence of 8.3% (1/12; 95% CI 0.4–35.4) and 11.2% (10/89; 95% CI 6.2–19.5), respectively (Table 2).

## OBI in the jaundiced Kenyan cohort; association of anti-HBc positivity and OBI

Anti-HBc antibody positivity was found in 56.9% (37/65; 95% CI 44.8–68.2) HBsAg negative specimens from individuals presenting with jaundice at a medical facility (S1C Fig, Table 2). Twenty samples (6 anti-HBc negative and 14 anti-HBc positive) tested positive for HBV DNA, resulting in 30.8% (95% CI 20.9 to 42.8) OBI positivity (Table 2). A significant association was observed between anti-HBc antibody positivity and OBI among all high-risk cohorts (Fisher's exact test; $P = 0.0153$) with an increase in significance observed when only the MSM-SW and jaundiced cohorts were considered (Fisher's exact test; $P = 0.007$).

## Associations with HBV DNA positivity among Kenyan MSM-SW cohort

HIV-reactivity results of MSM-SW and non-MSM individuals were determined during the original cohort study [20]. There was no significant association (Fisher's exact test) between MSM-SW HIV positivity and HBV DNA or anti-HBc positivity observed. All HBV DNA positivity was observed in unvaccinated men, other than 3 cases of OBI and 2 cases of HBsAg positive chronic infection in vaccinated HIV negative MSM-SW. All MSM-SW specimens tested negative for antibody to HCV, while HIV co-infection was present in 58.8% (10/17; 95% CI 36.0–78.4) of HBV DNA positive MSM-SW participants (S1A Fig). Association between HBV

**Table 3. Associations with HBV DNA positivity among MSM-SW cohort participants.**

| Variable | HBV DNA positive[a] | HBV DNA negative | P Value[b] |
|---|---|---|---|
| Marital status (single) | 14/16 (87.5%) | 54/75 (72.0%) | 0.2253 |
| Education | | | |
| Primary or less | 1/15 (6.7%) | 7/68 (10.3%) | 0.8985 (Chi-square) |
| Secondary | 7/15 (46.7%) | 29/68 (42.6%) | |
| Postsecondary | 7/15 (46.7%) | 32/68 (47.1%) | |
| Antiretroviral treatment | 12/15 (80.0%) | 26/68 (38.2%) | **0.0042** |
| Oral sex | | | |
| Often-Always[c] with a regular partner | 3/15 (20%) | 11/66 (16.7%) | 0.7165 |
| Often-Always with a casual partner | 2/11 (18.2%) | 8/54 (14.8%) | 0.6732 |
| Insertive anal sex | | | |
| Often-Always with a regular partner | 8/15 (53.3%) | 37/66 (56.1%) | >0.9999 |
| Often-Always with a casual partner | 5/12 (41.7%) | 28/53 (52.8%) | 0.5372 |
| Receptive anal sex | | | |
| Often-Always with a regular partner | 7/15 (46.7%) | 18/66 (27.3%) | 0.2140 |
| Often-Always with a casual partner | 7/12 (58.3%) | 16/52 (30.8%) | 0.1965 |

[a]Includes both HBsAg positive chronic and OBI DNA positive MSM-SW men. Not all participants in the original study cohort answered all interview questions.

[b]Fisher's exact test unless otherwise indicated.

[c]Often-Always is in comparison to Never-Sometimes. Bold indicates a P value <0.05.

DNA positivity and demographic, treatment or behavioural characteristics of MSM-SW cohort participants, determined during the original cohort study, was investigated (Table 3). The mean age of MSM-SW men was identical regardless of HBV DNA positivity (28 years), with similar median ages among the two groups (HBV DNA positive, 26 years; HBV DNA negative 25.5 years). Similarly, there were no significant associations observed with HBV DNA positivity, other than an association with the participant being treated with HIV antiretroviral therapy (Table 3).

## Phylogenetic analysis of HBV DNA positive samples

Following real-time PCR, specimens positive for HBV DNA (n = 38) underwent nested PCR for sequence analysis. Twenty-four specimens (S1 Table) were nested PCR positive and had sufficiently long sequence (at least 327 bp) for phylogenetic analysis. Seventeen OBI sequences from 1 non-MSM, 3 MSM-SW and 13 jaundiced participants, as well as 7 sequences from HBsAg positive MSM-SW specimens were aligned with GenBank reference sequences representing HBV subgenotypes, including 42 Kenyan HBV reference sequences (S1 Table), and were subjected to maximum likelihood phylogenetic analysis. All negative extraction and amplification controls were consistently negative, indicating control of possible environmental contamination. All study sequences were determined to be genotype A (Fig 1; GenBank accession no. MK487133—MK487155, MN972524). The MSM-SW sequences primarily clustered together, including with the single non-MSM OBI sequence (Cluster 1) and showed complete sequence identity over 327 nucleotides. Sequences from jaundiced patients did not cluster, although a second smaller cluster (Cluster 2) within the phylogenetic tree, comprised of a mixture of sequences from the MSM-SW and jaundiced cohorts, as well as several reference sequences from Kenya, showed complete sequence identity over 327 nucleotides. Definitive classification of the subgenotype of each sequence could not be determined due to subgenomic

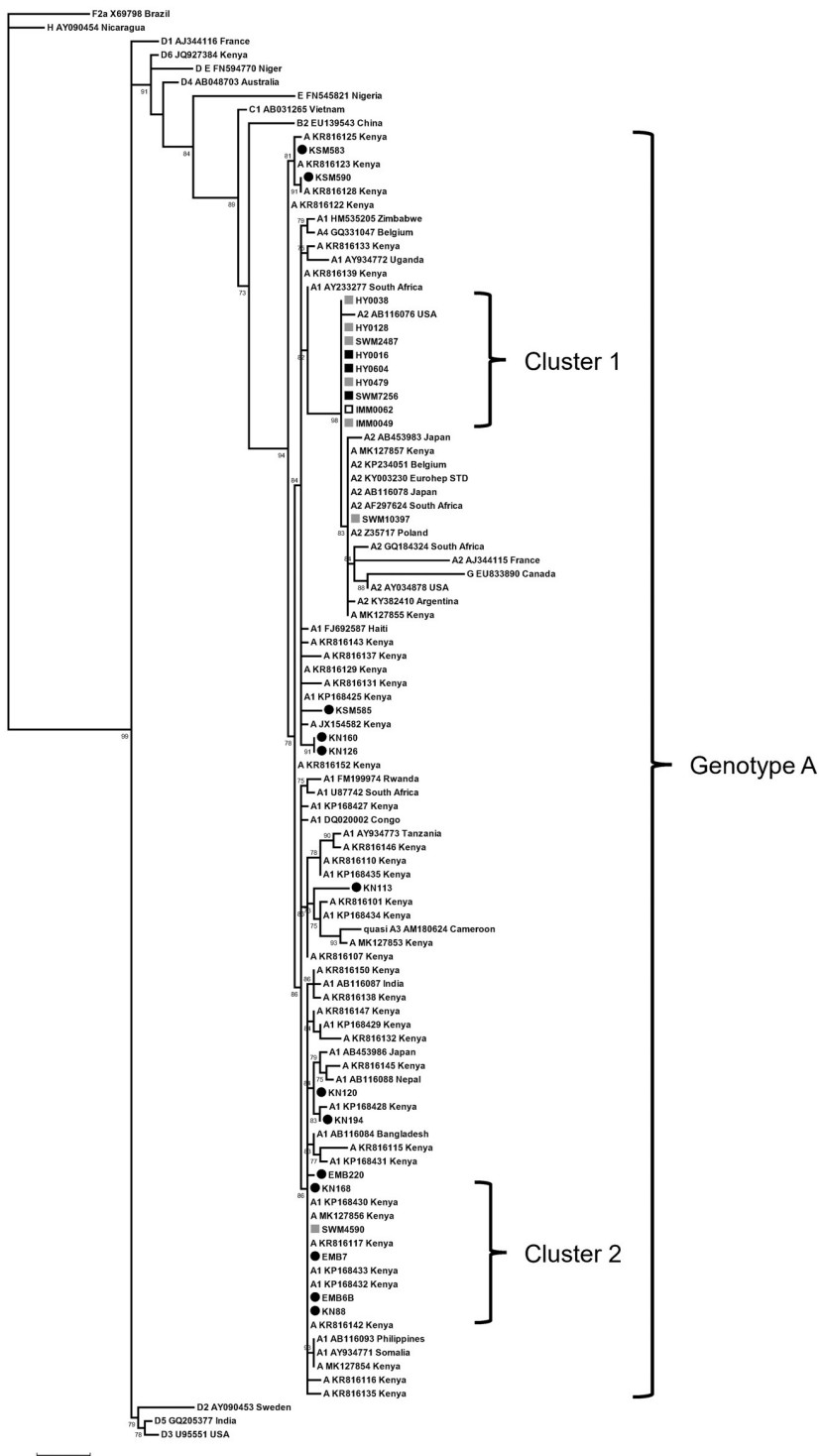

**Fig 1. Phylogenetic analysis of the HBsAg-coding region sequence (nt 458–784) from jaundiced OBI, non-MSM OBI, and MSM-SW OBI and HBsAg positive participants.** Twenty-four trimmed sequences (327 bp) from 17 OBI (comprising sequences from 1 non-MSM, 3 MSM-SW and 13 jaundiced participants) and 7 HBsAg positive MSM-SW specimens were aligned with GenBank reference sequences and analyzed by maximum likelihood method, using the most fit GTR+γ+I substitution model with the approximate likelihood ratio test for branch support statistics. Reference sequences are shown as a genotype or subgenotype followed by the GenBank accession number and country of origin if available. Two clusters are indicated. Branch support >70% is shown. The ruler shows the branch length for a pairwise distance equal to 0.05. Filled square, MSM-SW OBI sequences; open square, non-MSM OBI sequence; filled circle, jaundiced OBI sequences; grey square, MSM-SW HBsAg positive sequences.

region analysis [30]. However, sequences from jaundiced patients appeared to be more similar to subgenotype A1 based on tree topology. MSM-SW OBI sequences were observed exclusively within Cluster 1, which clustered most closely with subgenotype A2 reference sequences with 98% branch support (Fig 1). Mutations associated with immune escape or impaired HBsAg or virion secretion [29] were infrequently observed within amino acid alignments of OBI sequences. Only three mutations were observed within the HBsAg major hydrophilic region (amino acids 99 to 160) of three specimens from jaundiced individuals; T116N, T118A (both observed with KN126 and KN160; accession numbers MK487143 and MK487142, respectively) and D144E (KN113; accession number MK487153). No known nucleos(t)ide analog resistance mutations were observed within amino acid alignments of the polymerase reading frame (rtI111 to rtM218) within the study sequences.

## Discussion

Africa is an intermediate endemic region for hepatitis B infection, with an estimated prevalence of 6.1% [6]. An increased probability of infection is associated with certain populations at high risk, such as female sex workers [31], MSM [32, 33], or patients presenting for medical care with jaundice [19, 34, 35]. The findings of this study describe the high prevalence of HBsAg positivity in MSM-SW at 10.1%, and provide the first description of OBI in Kenya. None of the MSM-SW participants tested seropositive for HCV, comparable to other investigations describing a low HCV prevalence in Africa [19, 36, 37].

OBI is defined as the presence of HBV DNA in the liver or serum in the absence of detectable HBsAg, either in the presence or absence of other serological markers of HBV infection or exposure [38]. OBI is most often associated with very low quantitative HBV DNA levels (<200 IU/mL) [2] in the patient serum, consistent with a subclinical infection in which HBsAg levels may have fallen below the limit of diagnostic detection [39]. Although mutations within the HBsAg coding region which affect diagnostic detection are also a cause of "false" OBI [1], these mutations were infrequently observed among OBI-defined HBV sequences in this study. The risk of OBI is dependent on various factors including population prevalence and HIV co-infection; however, OBI detection is also dependent on test sensitivity and specificity [40]. Expert recommendations for OBI detection state that very sensitive DNA amplification procedures (nested PCR or real-time PCR) should be employed using multiple primer sets specific for highly conserved regions of the genome [1]. The present study followed this recommendation and determined OBI by a positive signal within at least two different genomic regions by real-time PCR, followed by nested PCR to obtain sequence information. Consistent with the intermediate endemicity of HBV in Africa, high rates (>6%) of OBI have been reported, in Nigerian blood donors [10, 41] and HIV positive patients in Botswana [14] or Cameroon [13]. Therefore the OBI rate of 18.7% (31/166; 95% CI 13.5–25.3) among HBsAg negative individuals analyzed in the present study would be expected in keeping with prior studies conducted in sub-Saharan Africa, and considering the population tested comprised individuals at high risk of HBV infection.

A difference in OBI prevalence among the MSM-SW and jaundiced cohorts was observed, such that the jaundiced patient group exhibited a nearly 3-fold higher rate of OBI (20/65; 30.8%, 95% CI 20.9–42.8) compared to the MSM-SW group (10/89; 11.2%, 95% CI 6.2–19.5). Risk factors associated with transmission may be an influence on this proportional difference. Universal HBV vaccination in Kenya was initiated in 2001 with provision of the pentavalent (DPT-HepB-Hib) vaccine at 6, 10, and 14 weeks of age [42]. Prior to vaccination program implementation, high rates of chronic hepatitis B infection in Africa likely resulted from transmission during childhood before the age of 5 years [6], leading to a higher risk of developing

chronic infection (30% to 50% in childhood). Despite the increased risk of HBV transmission with high-risk sexual activity among the MSM-SW cohort, the risk of developing chronic infection in adulthood is decreased (5 to 15%) compared to infection in childhood. Thus, although both groups are drawn from the general population, symptomatic patients are more likely to harbour overt and subclinical infection. HBsAg prevalence in the original jaundiced cohort was 50.6% [19], so the overall risk of HBV infection appears to be higher in patients presenting with jaundice for medical care compared to MSM-SW individuals. A rise in OBI prevalence has been associated with increasing age among certain populations [43], which may in part explain the 3-fold increased rate of OBI among the jaundiced cohort, as the mean age of individuals in this group was approximately a decade greater (37.8 years) than the mean age of participants of the MSM-SW cohort (28.6 years). There were no significant associations between HBV DNA positivity and demographic, treatment or behavioural characteristics of MSM-SW cohort participants other than an association with HIV antiretroviral therapy ($P$ = 0.0042; Fisher's exact test). The lack of significant association with HBV DNA positivity may be due to the very small numbers of participants with available data. The association between HIV treatment and detectable HBV DNA may be due to lower CD4 cell counts, increased HIV viral load ($\geq$ 300 copies/mL), or less time since antiretroviral therapy initiation within this population, as was observed in a study of Cameroonian HBV/HIV co-infected persons [44]. Furthermore, individuals may more readily accept and comply with treatment if undiagnosed HBV co-infection results in increased symptoms or adverse clinical outcomes.

All OBI participants as well as HBsAg positive MSM-SW individuals, were determined to be infected with HBV genotype A, which has been shown to be highly prevalent in high-risk groups in Kenya, based on reports from jaundiced [19], HIV-coinfected [45, 46] or PWID [47] cohorts. However, there were unexpected differences in phylogenetic clustering of OBI sequences among MSM-SW and jaundiced patients. All sequences from MSM-SW OBI and most from HBsAg positive MSM-SW persons clustered with GenBank subgenotype A2 reference sequences, with branch support of 98%, whereas sequences from jaundiced OBI were dispersed throughout the tree among subgenotype A1 reference sequences. HBV subgenotype A1 is highly predominant in Africa [48], including Kenya [18, 19] while subgenotype A2 is normally observed outside of Africa. Strikingly, all MSM-SW sequences clustering with A2 reference sequences showed complete sequence identity over 327 bp of alignment. This suggests that a common source or network transmission may be occurring with involvement of a non-African HBV transmission source within the MSM-SW cohort. However, it is important to consider that the relatively short HBsAg-coding region alignment used in this analysis is fairly well conserved and so inadvertent clustering may occur, as is evidenced by sequence identity of several study specimens with GenBank reference sequences, including several originating from Kenya. Thus phylogenetic analysis with full genome sequence is required to completely delineate HBV case relationships and resolve subgenotype classification.

A lack of mutations associated with resistance to antiviral therapy among the viral sequences investigated in this study was observed. Participants in all 3 cohorts were not known to have been treated with antiviral therapy for HBV, and thus would be considered treatment naïve, although 83 MSM-SW reported being treated for HIV infection and we cannot exclude the possibility that other participants may have been exposed to antiviral therapy in the past. Several HIV therapies are effective against HBV, and it has been suggested that transmission of drug-resistant HBV strains is likely occurring within Africa [28], partly due to the early widespread use of lamivudine as a treatment for HIV. Although tenofovir disoproxil fumarate (TDF) is the currently recommended first-line treatment for HBV/HIV coinfection in Africa, depending on the dosage, patient compliance, and prior use of other monotherapies, such as lamivudine, the risk of developing HBV resistance mutations resulting in viral

breakthrough increases in HBV/HIV coinfection [49]. Furthermore, cross-resistance among therapies is common, thus the use of therapies having a high genetic barrier to resistance, such as TDF, is crucial for controlling HBV viral breakthrough. At present, the availability of TDF for HBV mono-infection in Africa is not reliable [28].

The HBsAg-coding region mutations associated with immune escape [28, 50], T116N, T118A and D144E, were observed in several participants within the jaundiced cohort. The vaccination status of jaundiced participants was not known, nor were other possible sources of immune pressure that could result in HBsAg mutations. As the reverse transcriptase region of the HBV polymerase gene overlaps with the HBsAg-coding region, random mutations in either open reading frame may lead to changes in HBsAg expression, release or immunogenicity, potentially affecting vaccine response, HBsAg detection, and increased fitness of drug-resistant strains [51]. The study data does not provide sufficient evidence to understand whether drug-resistant or immune escape mutations pose an increased risk to high-risk individuals in Kenya; however, it would be prudent to diagnose and monitor HBV infection in all HIV-positive individuals prior to treatment initiation [49].

## Limitations

There were several limitations within the study which may have affected results and interpretation. Specimens were selected from archived sera related to studies of MSM-SW and jaundiced patients in Kenya based on a remaining sufficient volume for the study of OBI, which may have introduced selection bias. Insufficient participant numbers were available for statistical analysis, such that only major associations could be observed. Thus although a significant association among OBI and mutations or behavioral variables was not observed in this study, it may exist in a study with appropriate power. Although specimens were tested by a validated real-time PCR method targeting 3 unique HBV genome regions, specimens testing negative were not re-extracted or re-tested using a different approach. Thus, there is a possibility that prevalence estimates are under-reported.

## Conclusions/implications

OBI was determined to be highly prevalent among Kenyan individuals at high risk of HBV infection, including jaundiced patients presenting for medical care (30.8%; 95% CI 20.9–42.8), male sex workers (11.2%; 95% CI 6.2–19.5) and non-MSM men at high risk of HIV infection (8.3%; 95% CI 0.4–35.4). HBsAg positive chronic infection was also observed at a similar high rate (10.1%; 95% CI 5.6–17.6) in MSM-SW. A majority (58.8%; 95% CI 36.0–78.4) of MSM-SW were found to be co-infected with HIV, emphasizing the suggested practice of screening all HIV positive patients for HBV infection to establish appropriate treatment and patient management [49]. As OBI was significantly associated with anti-HBc positive status among patients at high risk for HBV infection, suspect cases of viral hepatitis should include screening by anti-HBc and HBV DNA, using highly sensitive methods to detect the low HBV DNA levels associated with OBI [2]. Due to the high risk of HBV infection in sex workers, screening and HBV vaccination of susceptible individuals should be implemented in MSM-SW cohorts to prevent the transmission of HBV and allow linkage to care for those infected [6]. The HBV genotype distribution differed among jaundiced and MSM-SW participants consistent with differing risk factors for transmission and suggests a possible common non-African source circulating among the cohort, resulting in overt and subclinical infection. As OBI remains a risk for HBV transmission and the development of severe liver disease [3], acknowledging the burden of infection represented by OBI in Kenya is necessary to develop national prevention and control measures.

## Supporting information

**S1 Fig. Flow charts of study specimen HBV testing and results.** Shaded boxes denote OBI positive results. The total number of specimen results or specimens tested is noted within the box. MSM-SW specimens (A), non-MSM specimens (B), jaundiced patient specimens (C). HIV-reactivity results were determined during the original cohort study [20]. neg, Negative; pos, Positive.
(PDF)

**S1 Table. HBsAg-coding region sequences included in phylogenetic analysis, by cohort or reference.**
(PDF)

## Acknowledgments

Ms. Jepkemei was a recipient of a Queen Elizabeth II Diamond Jubilee Scholarship for the duration of the study.

## Author Contributions

**Conceptualization:** Kiptoon Beatrice Jepkemei, Lyle R. McKinnon, Anton Andonov, Carla Osiowy.

**Data curation:** Missiani Ochwoto, Henok Gebrebrhan, Lyle R. McKinnon, Julius Oyugi, Joshua Kimani, Carla Osiowy.

**Formal analysis:** Kiptoon Beatrice Jepkemei, Carla Osiowy.

**Investigation:** Kiptoon Beatrice Jepkemei, Missiani Ochwoto, Jacqueline Day, Henok Gebrebrhan.

**Project administration:** George Gachara, Carla Osiowy.

**Resources:** Lyle R. McKinnon, Julius Oyugi, Joshua Kimani, Elijah Maritim Songok, Carla Osiowy.

**Supervision:** Ken Swidinsky, Carla Osiowy.

**Validation:** Kiptoon Beatrice Jepkemei, Ken Swidinsky, Jacqueline Day, Anton Andonov.

**Visualization:** Kiptoon Beatrice Jepkemei.

**Writing – original draft:** Carla Osiowy.

**Writing – review & editing:** Kiptoon Beatrice Jepkemei, Missiani Ochwoto, Ken Swidinsky, Jacqueline Day, Henok Gebrebrhan, Lyle R. McKinnon, Anton Andonov, Julius Oyugi, Joshua Kimani, George Gachara, Elijah Maritim Songok, Carla Osiowy.

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
