## [Decision Letter · Decision Letter 0]

30 Mar 2020

PONE-D-20-02941

Characterization of occult hepatitis B in high-risk populations in Kenya

PLOS ONE

Dear Dr. Osiowy,

Thank you for submitting your manuscript to PLOS ONE. After careful consideration, we feel that it has merit but does not fully meet PLOS ONE’s publication criteria as it currently stands. Therefore, we invite you to submit a revised version of the manuscript that addresses the points raised during the review process.

We would appreciate receiving your revised manuscript by May 14 2020 11:59PM. To enhance the reproducibility of your results, we recommend that if applicable you deposit your laboratory protocols in protocols.io, where a protocol can be assigned its own identifier (DOI) such that it can be cited independently in the future. For instructions see: http://journals.plos.org/plosone/s/submission-guidelines#loc-laboratory-protocols

We look forward to receiving your revised manuscript.

Kind regards,

Jason Blackard, PhD

Academic Editor

PLOS ONE

Additional Editor Comments (if provided):

This is a cross-sectional (convenience sampling) study of occult HBV infection in Kenya.  Overall, the methods and results are clear and the writing is concise.  A few minor revisions would further strengthen this manuscript, including:

The prevalence of occult HBV observed (18.7%) was about what would be expected based on other studies conducted in sub-Saharan Africa.  

Were samples that were found negative for HBV DNA tested a second time using a new extraction or different PCR approach?

Were ALT values available for any of the sub-populations studied?  What is the associated between detection of HBV DNA and ALT levels?

It is unclear what specific mutations were screened for . . . perhaps list them explicitly in the methods?  Similarly, what specific resistance mutations were screened for?

The discussion should include a statement that the statistical analyses were almost certainly not powered to find anything more than the largest associations.  No significant associations does not mean they don’t exist if the study was not powered appropriately.

Can additional sequences from Kenya be included in the phylogenetic analysis?

Journal Requirements:

2. Thank you for inlcuding your ethics statement; "Ethical approval for collection and investigation of specimens from jaundiced patients seeking medical care was obtained from the Kenya Medical Research Institute’s National Ethical Review committee, approval number SSC 2436. Informed consent was given by each participant or guardian through a signed consent form prior to drawing a blood sample and obtaining demographic information. Ethical approval for collection and investigation of specimens from MSM and non-MSM participants was obtained through approval of institutional review boards at the Kenyatta National Hospital ERC and the University of Manitoba with the collection of signed consent. "

Please amend your current ethics statement to confirm that your named institutional review board or ethics committee specifically approved this study.

Reviewers' comments:

Reviewer's Responses to Questions

**Comments to the Author**

1. Is the manuscript technically sound, and do the data support the conclusions?

Reviewer #1: Partly

Reviewer #2: Partly

2. Has the statistical analysis been performed appropriately and rigorously? 

Reviewer #1: No

Reviewer #2: I Don't Know

3. Have the authors made all data underlying the findings in their manuscript fully available?

Reviewer #1: Yes

Reviewer #2: Yes

4. Is the manuscript presented in an intelligible fashion and written in standard English?

Reviewer #1: Yes

Reviewer #2: No

5. Review Comments to the Author

Reviewer #1: This is an interesting and relevant piece of research on the phenomenon of occult HBV in high-risk populations in Kenya. As occult HBV can lead to significant morbidity, particularly in otherwise vulnerable populations, understanding of its transmission and prevalence remains useful, even in an era of near-universal vaccination.

However, there are some challenges with this manuscript that require additional focus prior to publication.

First, please ensure an appropriate definition of "endemic." The paper cited for that definition suggests >8% prevalence, yet multiple times in the manuscript HBV is reported as "endemic" in sub-Saharan Africa and Kenya at a prevalence of 6%.

Second, the patient populations selected are clear in their characterization, but these appear to be convenience cohorts from prior studies. Rather than using Figure 1 to show serological marker outcomes (which can be fully displayed in Table 2 instead), perhaps Figure 1 can show what cohort is what. How many patients were MSM-SW in total? Were all serum samples in that cohort tested? What about non-MSM with known HIV+ partners, and those with jaundice? Were these all previously identified for a study, and again, were all patients included in this retrospective analysis? If the answer to this is "yes", this should be clarified. If "no", please explain how samples were selected.

Third, why were only the first cohort tested for HCV (MSM-SW)?

Fourth, sample size does limit statistical analysis but it would be helpful to have a short paragraph on what stats were run and why (Fisher's Exact was the only test mentioned, I believe).

Fifth, the association with ART deserves a bit more consideration. Please put into context with current literature. Likewise, mutations causing HBV to escape detection, particularly in the setting of HIV, can be better described. See some of the work stemming from the Sherman lab.

Finally, conclusions should discuss implications of this work. Should patients presenting with jaundice be screened for HBV DNA, regardless of other markers? Same for MSM-SW? What are the challenges with that approach?

Reviewer #2: The study determined the prevalence and molecular characteristics of occult hepatitis B infections in populations at high risk of occult HBV infections in Kenya. Hepatitis B markers were screened for in 99 male sex workers having sex with men, 13 Non –MSM having HIV positive partners and 65 patients presenting with jaundice. The study reported a high occult HBV prevalence of 31 (18.7%). The predominant subgenotype circulating amongst jaundiced patients was A1 while for MSM-SW it was A2.

The study addressed important gaps in an under researched field. Occult HBV is becoming clinically relevant and more data is needed in this area. The study was well conducted. However, the definition of HBsAg positive chronic HBV is missing leading to confusion of some of the results. The study recommendations/implications are missing in both the abstract and the conclusion section. The study did not discuss any limitations.

ABSTRACT

1. Line 23: Correct the occult hepatitis B infection definition to presence of hepatitis B virus DNA.

2. Line 31: Write anti-HBc in full since it is the first mention

3. Line 33: Write HCV in full for the same reason as above

4. Line 33 and 36. Include confidence intervals in prevalence data. Here and elsewhere

5. Line 36: Please confirm HBsAg positive prevalence. In Figure 1A there are 10 HBsAg MSM-SW

6. Lines 31 and 32 implies that all samples were tested for HBsAg in this study but the 65 jaundiced patients were pre-screened as shown in Figure 1C. Please rephrase.

7. The abstract lacks study recommendation or implications of the study results.

INTRODUCTION

The introduction is well written.

MATERIALS AND METHODS

8. Line 95: Company address for Qiagen is missing

9. Line 115: Inconsistency , use of occult HBV versus OBI here and elsewhere.

10. Line 119: ThermoFisher Scientific address missing.

RESULTS

11. Line 154-155: How was chronic HBV infection defined in this study? There were 10 HBsAg MSM-SW but the prevalence of HBsAg positive chronic HBV infection is stated as 7/99.

12. Some of the results are presented in a confusing manner. It would be better to present all prevalence data for the high risk groups and then the associations between demographics and other parameters with OBI later. That is finish discussing table 2 and then move to table 3 instead of moving back and forth between the 2 tables.

13. Line 193. Please include the high risk group breakdown of the 24 samples which were genotyped. The information in lines 220-221 should be within text, not only in legend.

DISCUSSION

14. Line 233 , 234 and 239 the abbreviations (MSM-SW, MSM, OBI) have been previously defined in the manuscript and does not need to be defined again here.

15. The study did not describe any limitations

CONCLUSION

The recommendations or implications of the study results is missing

6. PLOS authors have the option to publish the peer review history of their article (what does this mean?). If published, this will include your full peer review and any attached files.

Reviewer #1: No

Reviewer #2: No

---

## [Author Response · Author response to Decision Letter 0]

22 Apr 2020

A response to reviewers file has been uploaded. The contents of the file are copied below:

Response to Editor and Reviewers

We thank the Academic Editor and Reviewers for their helpful comments and suggestions to correct, clarify and improve the manuscript. All page and line descriptions of revisions refer to the unmarked “Manuscript” file. Please see responses to the comments as follows.

Academic Editor Comments:

The prevalence of occult HBV observed (18.7%) was about what would be expected based on other studies conducted in sub-Saharan Africa.

As suggested, we have included a statement stating that the prevalence observed “…would be expected in keeping with prior studies conducted in sub-Saharan Africa…” (page 16, lines 284-285).

Were samples that were found negative for HBV DNA tested a second time using a new extraction or different PCR approach?

All specimens were tested using a highly sensitive real-time PCR assay targeting 3 different HBV genomic regions, with a positive result determined by at least 2 regions having Ct values below a specified cut-off. Therefore, samples negative by real-time PCR were not tested or extracted a second time. The possibility that this lack of secondary testing may have resulted in under-reporting of the prevalence estimate was added as a limitation to the revised manuscript (page 20, lines 371-374).

Were ALT values available for any of the sub-populations studied? What is the associated between detection of HBV DNA and ALT levels?

Unfortunately ALT levels were not available for the sub-populations studied. The ALT levels of HBV-infected individuals will fluctuate depending on the phase of infection of the patient. For example, ALT levels will be persistently normal during the HBeAg-positive chronic infection or ‘immunotolerant’ phase of infection, in which HBV DNA levels are very high. Yet once HBV viral load levels start to decrease in the HBeAg-positive chronic hepatitis phase of infection, ALT levels will fluctuate and flare above normal. This is due to the host immune response recognizing the virus and destroying infected hepatocytes, resulting in increases in detectable liver enzymes. A sentence was added to the Materials and Methods section (Page 4, lines 79-81) to state that ALT levels were not available for analysis.

It is unclear what specific mutations were screened for . . . perhaps list them explicitly in the methods? Similarly, what specific resistance mutations were screened for?

Thank you for this suggestion to help improve the manuscript. The specific amino acid mutations that were screened for both the HBsAg (immune escape) and polymerase reverse transcriptase region (drug resistance) was added to the revised Materials and Methods section along with citations describing the mutations (page 8, lines 132-138).

The discussion should include a statement that the statistical analyses were almost certainly not powered to find anything more than the largest associations. No significant associations does not mean they don’t exist if the study was not powered appropriately.

We appreciate this excellent suggestion. The following statement was added to the Limitations section (page 20, lines 368-371): “Insufficient participant numbers were available for statistical analysis, such that only major associations could be observed. Thus although a significant association among OBI and mutations or behavioral variables was not observed in this study, they may exist in a study with appropriate power.”

Can additional sequences from Kenya be included in the phylogenetic analysis?

Yes, we have re-analyzed the phylogenetic tree, adding an additional 31 genotype A GenBank sequences originating from Kenya above the original 11 Kenyan reference sequences. All of the sequences within the tree have been tabulated by Kenyan GenBank reference, non-Kenyan GenBank reference, MSM-SW OBI, MSM-SW HBsAg positive, non-MSM OBI, and Jaundiced OBI (S1 Table and Fig 1).

Journal Requirements: Please amend your current ethics statement to confirm that your named institutional review board or ethics committee specifically approved this study. Once you have amended this/these statement(s) in the Methods section of the manuscript, please add the same text to the “Ethics Statement” field of the submission form (via “Edit Submission”).

The Ethics statement in the manuscript and the submission form has been revised to include the statement: “Both institutional research ethics boards approved the investigation of infectious pathogens in consenting participants, including the detection and characterization of hepatitis B and hepatitis C viruses.” (Page 9, lines 152-154).

Reviewer #1 Comments:

First, please ensure an appropriate definition of "endemic." The paper cited for that definition suggests >8% prevalence, yet multiple times in the manuscript HBV is reported as "endemic" in sub-Saharan Africa and Kenya at a prevalence of 6%.

Thank you for this correction. This has been revised throughout the manuscript to state the prevalence in sub-Saharan Africa and Kenya meets intermediate endemicity levels, with the definition provided in the Introduction (2% to 7%; page 3, line 51).

Second, the patient populations selected are clear in their characterization, but these appear to be convenience cohorts from prior studies. Rather than using Figure 1 to show serological marker outcomes (which can be fully displayed in Table 2 instead), perhaps Figure 1 can show what cohort is what. How many patients were MSM-SW in total? Were all serum samples in that cohort tested? What about non-MSM with known HIV+ partners, and those with jaundice? Were these all previously identified for a study, and again, were all patients included in this retrospective analysis? If the answer to this is "yes", this should be clarified. If "no", please explain how samples were selected.

Thank you for the opportunity to clarify. The total numbers of each cohort in the original study were provided in the revised manuscript Materials and Methods Specimen Collection paragraph (page 4, lines 67 to 74 and page 5, lines 84 to 87). This added information provides the clarification that not all of the original study samples were tested for OBI (or HBsAg positivity for MSM-SW) due to sample exhaustion and that only specimens having sufficient serum volume for OBI investigation were selected. 

As Figure 1 was suggested to be redundant, it was moved to Supplementary Information (S1 Fig) to provide a complementary breakdown of samples and results to Table 2.

Third, why were only the first cohort tested for HCV (MSM-SW)?

The lack of anti-HCV testing in jaundiced individuals was explained in the revised Materials and Methods section (page 5, lines 87-88): “All jaundiced patients included in this study had been tested for antibody to hepatitis C virus (HCV) in the original study and were found to be negative [19].”

Fourth, sample size does limit statistical analysis but it would be helpful to have a short paragraph on what stats were run and why (Fisher's Exact was the only test mentioned, I believe).

Thank you for this opportunity to correct and improve the manuscript. The statistical analysis section of the Materials and Methods has been revised to describe the statistical tests used for all analyses, including confidence intervals (page 9, lines 157-162).

Fifth, the association with ART deserves a bit more consideration. Please put into context with current literature. Likewise, mutations causing HBV to escape detection, particularly in the setting of HIV, can be better described. See some of the work stemming from the Sherman lab.

As suggested, we have included a discussion regarding the association of HBV DNA with antiretroviral therapy in the Discussion (page 17, lines 310-315). An expanded discussion of polymerase and HBsAg mutations associated with drug resistance and immune escape, respectively, in the context of HIV infection has also been added to the Discussion (pages 18-19, lines 336-362), as suggested.

Finally, conclusions should discuss implications of this work. Should patients presenting with jaundice be screened for HBV DNA, regardless of other markers? Same for MSM-SW? What are the challenges with that approach?

We thank the reviewer for this comment to improve the manuscript. As suggested we have added a section at the end of the manuscript “Conclusions/Implications” (page 20-21, lines 376-394), which discusses suggested measures for screening, treatment and control of HBV infection in Kenya based on the data presented. Similar implications of the data were further provided during the discussion of drug resistance and immune escape (page 19, lines 347-350 and 359-362) and a sentence was added to the end of the Abstract (page 2, lines 38-41).

Reviewer #2 Comments:

However, the definition of HBsAg positive chronic HBV is missing leading to confusion of some of the results. 

We thank the reviewer very much for catching this glaring oversight! We had incorrectly calculated the prevalence of “chronic” HBV infection due to our focus on HBV DNA alone. However, by including a definition of chronic HBV infection (defined as HBsAg positivity; Page 9, line 167) the correct prevalence calculation, including all MSM-SW individuals positive for HBsAg, was provided (10/99; 10.1%, 95% CI 5.6-17.6). This correction was made throughout the manuscript, including the Abstract.

The study recommendations/implications are missing in both the abstract and the conclusion section. 

We thank the reviewer for this comment to improve the manuscript. As suggested we have added a section at the end of the manuscript “Conclusions/Implications” (page 20-21, lines 376-394), which discusses suggested measures for screening, treatment and control of HBV infection in Kenya based on the data presented. Similar implications of the data were further provided during the discussion of drug resistance and immune escape (page 19, lines 347-350 and 359-362) and a sentence was added to the end of the Abstract (page 2, lines 38-41).

The study did not discuss any limitations.

As suggested, a section on study limitations was added to the manuscript, expanding on comments and limitations noted by the academic editor and reviewers (page 20, lines 364-374).

ABSTRACT

1. Line 23: Correct the occult hepatitis B infection definition to presence of hepatitis B virus DNA.

We thank the reviewer for catching this oversight. The definition was corrected to state “Occult hepatitis B infection (OBI) is defined as the presence of hepatitis B virus (HBV) DNA in the liver or serum in the absence of detectable HBV surface antigen (HBsAg).” (page 2, lines 23-24). This was also corrected in the Discussion (page 15, lines 267-268).

2. Line 31: Write anti-HBc in full since it is the first mention

The line in the Abstract was changed to “…which were HBV core protein antibody positive.” (page 2, line 34) and the first mention of anti-HBc was written in full in the Introduction “…antibody to the core (anti-HBc) or surface (anti-HBs) proteins…” (page 3, lines 44-45).

3. Line 33: Write HCV in full for the same reason as above

The first instance of HCV was written in full in the Abstract (page 2, line 32) and the manuscript text (Materials and Methods; page 5, lines 87-88).

4. Line 33 and 36. Include confidence intervals in prevalence data. Here and elsewhere

We thank the reviewer for the opportunity to correct and improve the manuscript. Confidence intervals of prevalence estimates were calculated by computing the confidence interval of a proportion by the Wilson/Brown method (this statement was added to page 9, lines 160-161). Throughout the text, prevalence values were followed by the calculated 95% CI range.

5. Line 36: Please confirm HBsAg positive prevalence. In Figure 1A there are 10 HBsAg MSM-SW

As mentioned above, we had incorrectly calculated the prevalence of HBsAg positivity due to our focus on HBV DNA alone. The HBsAg positive prevalence for study MSM-SW participants was corrected (10/99; 10.1%, 95% CI 5.6-17.6). This correction was made throughout the manuscript, including the Abstract.

6. Lines 31 and 32 implies that all samples were tested for HBsAg in this study but the 65 jaundiced patients were pre-screened as shown in Figure 1C. Please rephrase.

Thank you for this comment. As suggested, the methods portion of the Abstract was re-phrased to correct and clarify this point: “Sera from two Nairobi cohorts, 99 male sex workers, primarily having sex with men (MSM-SW), and 13 non-MSM men having HIV-positive partners, as well as 65 HBsAg-negative patients presenting with jaundice at Kenyan medical facilities, were tested for HBV serological markers, including HBV DNA by real-time PCR.” (page 2, lines 27-31).

7. The abstract lacks study recommendation or implications of the study results.

As suggested, we have included a sentence at the end of the Abstract (page 2, lines 38-41) to provide recommended measures for screening, treatment and control of HBV infection in Kenya based on the data presented.

MATERIALS AND METHODS

8. Line 95: Company address for Qiagen is missing

The company address for Qiagen was added (page 5, line 101). 

9. Line 115: Inconsistency , use of occult HBV versus OBI here and elsewhere.

We have corrected the use of OBI throughout the manuscript, with only the initial mention of occult HBV fully spelled out in the manuscript text on page 3, lines 43-44. 

10. Line 119: ThermoFisher Scientific address missing.

The company address for ThermoFisher Scientific was added (page 7, line 125). 

RESULTS

11. Line 154-155: How was chronic HBV infection defined in this study? There were 10 HBsAg MSM-SW but the prevalence of HBsAg positive chronic HBV infection is stated as 7/99.

As mentioned above, we thank the reviewer very much for catching this glaring oversight! We had incorrectly calculated the prevalence of “chronic” HBV infection due to our focus on HBV DNA alone. However, by including a definition of chronic HBV infection (defined as HBsAg positivity; Page 9, line 167) the correct prevalence calculation, including all MSM-SW individuals positive for HBsAg, was provided (10/99; 10.1%, 95% CI 5.6-17.6). This correction was made throughout the manuscript, including the Abstract.

12. Some of the results are presented in a confusing manner. It would be better to present all prevalence data for the high risk groups and then the associations between demographics and other parameters with OBI later. That is finish discussing table 2 and then move to table 3 instead of moving back and forth between the 2 tables.

We thank the reviewer for this comment to improve the manuscript. As suggested, we have moved the results of OBI in the jaundiced Kenyan cohort and the association of anti-HBc positivity with OBI directly following the results of OBI in MSM-SW and non-MSM cohorts (Page 10-11, lines 186-192), prior to the mention of Table 3. 

13. Line 193. Please include the high risk group breakdown of the 24 samples which were genotyped. The information in lines 220-221 should be within text, not only in legend.

Thank you for this comment. As suggested, a list of all cohort study samples that were sequenced, as well as GenBank reference sequences that were included in phylogenetic analysis was included as a supplementary Table (S1 Table). This information was also added to the text (Page 13, lines 220-223).

DISCUSSION

14. Line 233 , 234 and 239 the abbreviations (MSM-SW, MSM, OBI) have been previously defined in the manuscript and does not need to be defined again here.

The abbreviation definitions have been removed. 

15. The study did not describe any limitations

Thank you. As mentioned above, a section on study limitations was added to the manuscript, expanding on comments and limitations noted by the academic editor and reviewers (page 20, lines 364-374).

CONCLUSION

The recommendations or implications of the study results is missing

We thank the reviewer for this comment to improve the manuscript. As suggested we have added a section at the end of the manuscript “Conclusions/Implications” (page 20-21, lines 376-394), which discusses suggested measures for screening, treatment and control of HBV infection in Kenya based on the data presented. Similar implications of the data were further provided during the discussion of drug resistance and immune escape (page 19, lines 347-350 and 359-362).

---

## [Decision Letter · Decision Letter 1]

12 May 2020

Characterization of occult hepatitis B in high-risk populations in Kenya

PONE-D-20-02941R1

Dear Dr. Osiowy,

We are pleased to inform you that your manuscript has been judged scientifically suitable for publication and will be formally accepted for publication once it complies with all outstanding technical requirements.

With kind regards,

Jason Blackard, PhD

Academic Editor

PLOS ONE

Additional Editor Comments (optional):

None

Reviewers' comments:

Reviewer's Responses to Questions

**Comments to the Author**

1. If the authors have adequately addressed your comments raised in a previous round of review and you feel that this manuscript is now acceptable for publication, you may indicate that here to bypass the “Comments to the Author” section, enter your conflict of interest statement in the “Confidential to Editor” section, and submit your "Accept" recommendation.

Reviewer #2: All comments have been addressed

2. Is the manuscript technically sound, and do the data support the conclusions?

Reviewer #2: Yes

3. Has the statistical analysis been performed appropriately and rigorously? 

Reviewer #2: Yes

4. Have the authors made all data underlying the findings in their manuscript fully available?

Reviewer #2: Yes

5. Is the manuscript presented in an intelligible fashion and written in standard English?

Reviewer #2: Yes

6. Review Comments to the Author

Reviewer #2: (No Response)

7. PLOS authors have the option to publish the peer review history of their article (what does this mean?). If published, this will include your full peer review and any attached files.

Reviewer #2: No

---

## [Editor Report · Acceptance letter]

15 May 2020

PONE-D-20-02941R1 

Characterization of occult hepatitis B in high-risk populations in Kenya 

Dear Dr. Osiowy:

I am pleased to inform you that your manuscript has been deemed suitable for publication in PLOS ONE. Congratulations! Your manuscript is now with our production department. 

With kind regards,

on behalf of

Dr. Jason Blackard 

Academic Editor

PLOS ONE